# Effects of Dietary Crude Protein Levels on Fecal Crude Protein, Amino Acids Flow Amount, Fecal and Ileal Microbial Amino Acids Composition and Amino Acid Digestibility in Growing Pigs

**DOI:** 10.3390/ani10112092

**Published:** 2020-11-11

**Authors:** Zhenguo Yang, Tianle He, Gifty Ziema Bumbie, Hong Hu, Qingju Chen, Changwen Lu, Zhiru Tang

**Affiliations:** Laboratory for Bio-feed and Molecular Nutrition, College of Animal Science and Technology, Southwest University, Chongqing 400715, China; 18894335705@163.com (T.H.); giftyziema@gmail.com (G.Z.B.); huhong1020@163.com (H.H.); cqj9517@163.com (Q.C.); luchangwen678@163.com (C.L.)

**Keywords:** dietary crude protein levels, essential amino acids, growing pigs, apparent digestibility, composition of fecal and ileal microbial amino acids

## Abstract

**Simple Summary:**

The purpose of this experiment was to evaluate a low protein corn-soybean meal-based diet with the same Lys, Met + Cys, Thr and Trp level as a high protein diet on fecal crude protein (CP), amino acid (AA) flow amount, AA digestibility and fecal and ileal microbial AA composition in growing pigs. Eighteen pigs with an initial body weight of (30 ± 1.35) kg were randomly divided into three groups, with six replicates in each group, and fed a corn-soybean meal-based diets with 12%, 15% and 18% CP levels, respectively. Our aim was to explain whether the addition of four crystalline essential AAs (EAA) to a low diet affected the digestibility of protein-bound NEAA (non-essential amino acid) and EAA and the composition of microbial AA in ileum and feces.

**Abstract:**

The purpose of this experiment was to evaluate the effects of low protein corn-soybean meal-based diets on fecal CP, amino acid (AA) flow amount, AA digestibility and fecal and ileal microbial AA composition in growing pigs. Eighteen pigs (initial body weight = 30 ± 1.35) were randomly divided into three groups and fed with basal diets with CP levels of 12%, 15% and 18%, respectively. The Lys, Met + Cys, Thr and Trp level in the 12% CP and 15% CP groups is the same as 18% CP group by the addition of four crystalline Lys, Met + Cys, Thr and Trp to the diet. The results showed that with the decrease of dietary CP level from 18% to 12%, the fecal total nitrogen (N), CP and total AA (TAA) flow amount decreased linearly (*p* < 0.05). Dry matter (DM) digestibility, CP digestibility, TAA digestibility, essential amino acid (EAA) digestibility and non-essential amino acid (NEAA) digestibility increased linearly with the decrease of dietary CP concentration from 18% to 12%. Compared with 18% CP group, the flow amount of Asp, Ser, Glu, Gly, Tyr, Val, Leu and Phe in feces of pigs in the 15% CP group and 12% CP group decreased significantly, while the flow amount of Arg in the 15% CP group was lower than that in the 18% CP group and 12% CP group. The fecal microbial N and AA of the 15% CP group were higher than those of the 18% CP and 12% CP groups. Fecal TAA flow amount decreased linearly with the decrease of the dietary CP levels from 18% to 12%. Fecal TAA and NEAA flow amount also decreased linearly with the decrease of dietary CP level from 18% to 12%. Except for Glu, Gly, Met, Tyr, Thr and Phe, there were significant differences among the three groups in the composition of 17 kinds of AAs in fecal microorganisms. Among the 17 AA compositions of ileal microorganisms, except Tyr and Lys, the other AAs were significantly different among the three groups (*p* < 0.05)

## 1. Introduction

In recent years, the environmental pollution caused by nitrogen emission has become more and more serious with the continuous expansion of livestock breeding scales and intensive degrees. At present, the nitrogen emission from livestock and poultry in China is about 500–600 million tons, of which monogastric animals (mainly pigs) account for about 60% of the total nitrogen emissions. At the same time, China is seriously short of protein resources. In 2019, China’s domestic soybeans were about 11 million tons, while imported soybeans were 95.53 million tons. Therefore, to improve protein utilization efficiency and reduce nitrogen emissions has become an urgent scientific problem to be solved.

Nitrogen emission reduction has been studied for a long time, and so far the following nitrogen emission reduction technologies have been formed: preparing dietary protein with ideal AA model [1]; low protein diet technology [2]; increasing the proportion of fermentable carbohydrates in diet [3]; adding feed additives such as enzymes, probiotics and organic acids [4]. Among them, the nitrogen emission reduction effect of low protein diet technology is the most obvious. According to the theory of protein-amino acid nutrition balance, a low protein diet is a diet that can reduce dietary protein levels and nitrogen emissions by adding appropriate kinds and quantities of synthetic AAs without affecting animal production performance and product quality. The practical significance of a low-protein diet is very significant. Studies have shown that for every 1% reduction of CP in pig diet, nitrogen emissions can be reduced by 8% to 10%, and ammonia concentration in pig houses can be reduced by more than 10% [5]. In the case of supplementation of important EAA, a reduction of 2% to 3% in dietary CP content did not affect the nitrogen deposition or growth performance of pigs [6,7], but the growth performance of pigs was usually negatively affected when the dietary CP content was more than 3% [5,8]. The increasingly serious environmental pollution forces researchers to fundamentally solve the protein and amino acid nutrition of pigs while reducing the nitrogen excretion of pig feces. It is pointed out that reducing the dietary CP of growing pigs can effectively reduce the nitrogen excretion of pig manure [9]. Reducing N excretion in swine manure can be effectively accomplished by reducing dietary CP intake. However, reducing the protein concentration of diets, with no supplemental crystalline AA, decreases growth performance and meat and carcass quality [9] and inhibits the activity of the mammalian target of rapamycin (mTOR) and s6 kinase 1 (S6K1), leading to the restriction of protein synthesis [10]. In low-protein diets, the absorption of protein-bound amino acids may be reduced, resulting in impaired animal growth performance if the dietary protein content cannot be further reduced, although the environmental benefits of low-protein diets are significant, the economic benefits are not obvious. Large-scale farms usually take economic benefits as the primary assessment index. Therefore, although the research history of low-protein diet has a history of more than 100 years, it has not been fully promoted in production [11]. In view of this, it is necessary to carry out in-depth study on the regulation mechanism of nitrogen emission reduction of livestock, identify the regulation targets, and develop a low-protein diet with a more efficient effect of nitrogen emission reduction.

The purpose of this experiment was to determine the effects of a low protein corn-soybean-based diet with the same Lys, Met + Cys, Thr and Trp level as high protein diet on fecal CP, AA flow, fecal CP, AA digestibility and fecal and ileal microbial AA composition in 30–60 kg growing pigs. Four kinds of crystalline AAs were added to the corn-soybean meal-based diet to reduce the dietary CP concentration to meet the needs of essential AA. In addition, the EAA/NEAA ratio (EAA/NEAA) decreased with the increase of dietary CP concentration. Our aim was to explain whether the addition of four crystalline EAAs to a low-CP diet affected the digestibility of protein-bound NEAA and EAA and the composition of microbial AA in ileum and feces.

## 2. Materials and Methods

### 2.1. Experimental Animals, Design and Diets

All experimental procedures in this study were approved by Sciences Animal Ethics Committee of Chinese Academy of Sciences (Hunan, China; Ethic approval number: SYXK 2014-0002). Eighteen crossbred barrows (Duroc × Landrace × Yorkshire, weighing on average 30 ± 1.35 kg) were randomly allotted to 3 experimental groups in accordance with dietary CP levels of corn-soybean meal-based diets: 12%, 15% and 18%, with six replicates each group and one piglet each replicate. Ingredients and nutrient contents of diets were given in Table 1 and the Lys, Met + Cys, Thr and Trp level in the 12% CP and 15% CP groups is the same as 18% CP group by the addition of four crystalline Lys, Met + Cys, Thr and Trp to diet. The trial lasted 30 days. All pigs were kept individually in stainless steel metabolism crates (1.5 m × 0.5 m × 0.8 m) in an ambient indoor temperature maintained approximately at 24 °C. Natural lighting was used during the experiment. All pig feeds are prepared according to various nutrient requirement recommended by the government (NRC, 2012). Feeding and watering was *ad libitum* throughout the trial.

### 2.2. Measurements and Sampling

On day 25 to day 30, six pigs each group were chosen to collect feces. Then the pigs were bled and anesthetized by intramuscularly injecting pentobarbital sodium (50 mg/kg·BW). Samples of terminal ileal digesta were collected and put into polythene bags for microbial or chemical analysis. The samples for microbe separation were stored at 4°C and for chemical analysis were stored at −20 °C.

### 2.3. Chemical Analysis

DM was determined by drying to a constant mass in a forced air oven at temperature maintained 95. The previous research methods were used to separate other digested samples by differential centrifugation [12,13]. Firstly, pooled digesta were centrifuged immediately (250× *g* for 15 min at 4 °C), to give a fraction expected to obtain porcine cells and food particles. Secondly, fractions were centrifuged at 14,500× *g* for 30 min at 4 °C, giving precipitates expected to contain microbial cells. N and ash in feed, feces and microbials were measured by the Leco total combustion method [14]. The AA in feed, feces and microorganisms was measured by a Hitachi Lmur8800 AA automatic analyzer (Hitachi, Ltd., Tokyo, Japan) [15]. Fecal and microbial protein N (PN) and non-protein N (NPN) were measured by the method of Warren et al. [13].

### 2.4. Data Treatment and Statistical Analyses

Acid insoluble ash (AIA), a kind of endogenous indicator was measured to estimate the apparent digestibility of DM, CP and TAA in feces. The following is the relevant formula: 1-bc/ad. where a denotes the content of DM, CP or AA in diets (%); b denotes the content of DM, CP or AA in feces (%); c denotes the content of AIA in diet (%); d denotes the content of AIA in feces (%). AA compositions (%) appertained the content of single AA in TAA.

All results were expressed as mean ± standard error of mean (SEM). All data were subjected to a one-way analysis of variance using the general linear model (GLM) procedure of SAS 8.2 statistical software (SAS Institute, Inc. Cary, NC, USA) [16] according to a completely randomized one-factorial design. Differences among experimental groups were identified via the Student–Newman–Keuls (SNK) test. A difference was considered to be statistically significant when *p* < 0.05.

## 3. Results

### 3.1. Fecal N and CP Flow and Digestibility

Feces flow amount and apparent digestibility of CP and N of growing pigs were illustrated in Table 2. Feces CP, total N (TN) and NPN flow amounts in growing pigs were affected by the experimental treatments (*p* < 0.01), i.e., greatest in the 18% CP group, greater in the 15% CP group and lowest in the 12% CP group (*p* < 0.05). Fecal PN flow amounts was a highly significant difference among the three experimental groups (*p* < 0.01), i.e., lower in the 12% CP group than in the 15% and18% CP group (*p* < 0.05), but there were no significant differences in feces PN flow amount between the 15% CP group and the 18% CP group (*p* > 0.05). Feces Total microbial N and microbial PN flow amount were significantly different among the three experimental groups (*p* < 0.01), i.e., lowest in the 18% CP group, greater in the 12% CP group and greatest in the 15% CP group (*p* < 0.05). Feces apparent digestibility of CP, DM, TN, NPN and PN was also different significantly among the three experimental groups (*p* < 0.01), i.e., greatest in the 12% CP group, greater in the 15% CP group and lowest in the 18% CP group (*p* < 0.05). In addition, there were no significant discrepancies in fecal microbial NPN among the three experimental groups (*p* > 0.05).

### 3.2. Feces Total AA (TAA), Essential (EAA) Amino Acids and Non-Essential (NEAA) Flow Amount and Apparent Total Amino Acid Number (TAA) with or without Microbia

Feces TAA, EAA, NEAA flow amount and apparent digestibility with or without microbia of growing pigs were illustrated in Table 3. Feces TAA, EAA and NEAA flow amounts with or without microbia in growing pigs were affected by the experimental treatments (*p* < 0.01), i.e., greater in the 18% CP group than in the 12% CP and 15% CP groups (*p* < 0.05), but there were no significant differences in feces TAA, EAA and NEAA flow amounts with or without microbia in growing pigs between the 12% CP group and the 15% CP group (*p* > 0.05). Digestibility of TAA, EAA and NEAA in feces with microbia and digestibility of EAA in feces without microbial were affected by the experimental treatments (*p* < 0.01), i.e., greater in the 12% CP group than in the 15% CP and 18% CP groups (*p* < 0.05), but there were no significant differences in digestibility of TAA, EAA and NEAA in feces with microbia and digestibility of EAA in feces without microbia in growing pigs between the 12% CP group and the 15% CP group (*p* > 0.05). Digestibility of TAA and NEAA in feces without microbia in growing pigs was affected by the experimental treatments (*p* < 0.01), i.e., greatest in the 12% CP group, greater in the 15% CP group and lowest in the 18% CP group (*p* < 0.05). Feces EAA/NEAA flows with or without microbia in growing pigs were affected by the treatments (*p* < 0.01), i.e., greater in the 15% CP group than in the 12% CP group and the 18% CP group (*p* < 0.05). There were no significant differences in feces EAA/NEAA flow in growing pigs between the 12% CP group and the 18% CP group (*p* > 0.05).

### 3.3. Feces Amino Acids Flow Amount

Feces Ser, Glu, Ala, Gly, Tyr, Asp, Thr, Cys, Leu and Phe flow amounts were different among the three experimental groups (*p* < 0.01) as illustrated in Table 4, i.e., compared with the 15% CP and 12% CP group, it was higher in the 18% CP group. However, compared with the 15% CP group, it was higher in the 12% CP group (*p* < 0.05). There were no significant differences in feces Ser, Glu, Gly, Tyr, Asp, Thr, Leu and Phe flow amounts in growing pigs between the 15% CP and 12% CP groups *(p* > 0.05). There were no significant differences in feces Pro, Met and His flow amounts in growing pigs among the three treatments (*p* > 0.05). Feces Lys flow amounts were highly significant in their difference among the three experimental groups (*p* < 0.01), i.e., greater in the 15% CP group and the 18% CP group than in the 12% CP group (*p* < 0.01). Feces Leu flow amounts were different among three treatments (*p* < 0.05), i.e., greater in the 18% CP group than in the 12% CP group and the 15% CP group (*p* < 0.05). There were no significant differences in feces Lys flow amounts in growing pigs between the 15% CP group and the 18% CP group (*p* > 0.05). Feces Ala and Val flow amounts were also very significantly different among the three experimental groups (*p* < 0.01), i.e., greatest in the 18% CP group, greater in 12% CP group and lowest in 15% CP, (*p* < 0.05). Feces Arg flow amounts were different among the three experimental groups (*p* < 0.01), i.e., greater in the 12% CP group and the 18% CP group than in the 15% CP group (*p* < 0.05). There were no significant differences in feces Arg flow amounts in growing pigs between the 18% CP group and 12% CP group (*p* > 0.05).

### 3.4. AAs Digestibility

The AAs digestibility were illustrated in Table 5. Feces Ser, Glu, Gly, Pro, Tyr, Thr, Ile, Leu, Phe and Lys digestibility were very significantly different among the three experimental groups (*p* < 0.01), i.e., greater in the 12% CP group than in the 15% CP and 18% CP groups (*p* < 0.05), but there were no significant differences in feces Ser, Glu, Gly, Pro, Tyr, Thr, Ile, Leu, Phe and Lys digestibility in growing pigs between the 15% CP and 18% CP groups (*p* > 0.05). There were no significant differences in feces Arg flow in growing pigs among the three experimental groups (*p* > 0.05). Feces Ala and Val digestibility of pigs in the 15% CP group and the 12% CP group are higher than in the 18% CP group (*p* < 0.05), while there were no significant differences in feces Ala and Val digestibility in growing pigs between the 15% CP group and the 12% CP group (*p* > 0.05). Feces Cys and His digestibility of pigs in the 12% CP group and the 18% CP group are higher than in the 15% CP group (*p* < 0.05), but there were no significant differences in feces Cys and His digestibility in growing pigs between the 12% CP and 18% CP groups (*p* > 0.05). Feces Met digestibility was very significantly different among these three experimental groups (*p* < 0.01), i.e., greatest in 12% CP group, greater in 18% CP group and lowest in the 15% CP group (*p* < 0.05).

### 3.5. The Feces microbial AA Flow Amount

As shown in Table 6, feces microbial Ser flow amounts were different among the three experimental groups (*p* < 0.01), i.e., greatest in the 15% CP group, greater in the 18% CP group and lowest in the 12% CP group (*p* < 0.05). Feces microbial Gly, Ala, Pro, Thr, Ile, TAA and EAA flow amounts were different among the three experimental groups (*p* < 0.01), i.e., greatest in the 15% CP group, greater in 12% and lowest in the 18% CP group (*p* < 0.05). Feces microbial Tyr, Val, Met, Phe and NEAA flow amounts were different among the three experimental groups (*p* < 0.01), i.e., greater in the 15% CP group than in the 18% CP and 12% CP groups (*p* < 0.05), but there were no significant differences in feces microbial Tyr, Val, Met, Phe and NEAA flow in growing pigs between the 12% CP and 18% CP groups (*p* > 0.05). Feces microbial Asp flows were different among the three experimental groups (*p* < 0.01), i.e., greatest in the 18% CP group, greater in the 15% CP group and lowest in the 12% CP group (*p* < 0.05). Feces microbial Cys flow amounts were different among the three experimental groups (*p* < 0.01), i.e., greater in the 18% CP group than in the 15% CP and 12% CP groups (*p* < 0.05), while there were no significant differences in feces microbial Cys flow amounts in growing pigs between the 12% CP group and the 15% CP group (*p* > 0.05). Feces microbial Lys and His flow amounts were different among the three experimental groups (*p* < 0.01), i.e., greater in the 15% CP and 18% CP groups than in the 12% CP group (*p* < 0.05). There were no significant differences in feces microbial Lys and His flow in growing pigs between the 18% CP group and the 15% CP. Feces microbial Arg flow amounts were different among the three experimental groups (*p* < 0.01), i.e., greater in the 15% CP and 12% CP groups than in the 18% CP group (*p* < 0.05). There were no significant differences in feces microbial Cys flow amounts in growing pigs between the 12% CP group and the 15% CP group (*p* > 0.05). There were no significant differences in feces microbial Glu, Leu and EAA/NEAA flow amounts in growing pigs among the three experimental groups (*p* > 0.05).

### 3.6. The Feces Microbial AA Composition

Feces microbial Ser of AA compositions were different among the three experimental groups (*p* < 0.01), i.e., greatest in the 15% CP group, greater in the 18% CP group, lowest in the 12% CP group as illustrated in Table 7 (*p* < 0.05). Feces microbial Ala, Pro and Ile of AA compositions were different among the three experimental groups (*p* < 0.01), i.e., greater in the 12% CP and 15% CP groups than in the 18% CP (*p* < 0.05), but there were no significant differences in feces microbial Ala, Pro and Ile of AA compositions in growing pigs between the 12% CP and 15% CP groups (*p* > 0.05). Feces microbial Asp, Cys, Lys and His of AA compositions were different among the three experimental groups (*p* < 0.01), i.e., greater in the 18% CP group than in the 15% CP and 12% CP groups (*p* < 0.05), but there were no significant differences in feces microbial Asp, Cys, Lys and His of AA composition in growing pigs between the 12% CP and 15% CP groups (*p* > 0.05). Feces microbial Val of AA compositions were different among three treatments (*p* < 0.01), i.e., greater in the 15% CP group than in the 12% CP and 18% CP groups (*p* < 0.05), but there were no significant differences in microbial Val feces of AA compositions in growing pigs between 12% CP group and 18% CP group (*p* > 0.05).

Feces microbial Leu of AA compositions were different among the three experimental groups (*p* < 0.01), i.e., greater in the 12% CP and 18% CP groups than in the 15% CP group (*p* < 0.05), but there were no significant differences in feces microbial Leu of AA composition in growing pigs between the 12% CP and 18% CP groups (*p* > 0.05). Feces microbial Arg of AA compositions were different among the three experimental groups (*p* < 0.01), i.e., greatest in the 12% CP group, greater in the 15% CP and lowest in the 18% CP (*p* < 0.05). There were no significant differences in feces microbial Glu, Gly, Tyr, Thr, Met, Phe, NEAA, EAA and EAA/NEAA of AA composition in growing pigs among the three experimental groups (*p* > 0.05).

### 3.7. The Ileal Microbial AA Composition

Table 8 shows that the ileal microbial Ser, Pro, Val and Ile of AA compositions were different among the three experimental groups (*p* < 0.01), i.e., greater in the 12% CP and 15% CP group than in 18% CP group (*p* < 0.05), but there were no significant differences in feces microbial Ser, Pro, Val and Ile of AA composition in growing pigs between the 12% CP group and the 15% CP group (*p* > 0.05). The ileal microbial Glu of AA composition was different among the three experimental groups (*p* < 0.01), i.e., greater in the 18% CP group than in the 12% CP and 15% CP groups (*p* < 0.05), but there were no significant differences in feces microbial Glu of AA composition in growing pigs between the 12% CP group and the 15% CP group (*p* > 0.05). The ileal microbial Gly of AA composition was different among the three experimental groups (*p* < 0.01), i.e., greatest in the 15% CP group, greater in the 12% CP group and lowest in the 18% CP group (*p* < 0.05). The ileal microbial Ala of AA composition was different among the three experimental groups (*p* < 0.01), i.e., greatest in the 12% CP group, greater in the 15% CP group and lowest in the 18% CP group (*p* < 0.05). The ileal microbial Asp, Thr and His of AA composition was different among the three experimental groups (*p* < 0.01), i.e., greater in the 12% CP and 18% CP groups than in the 15% CP group (*p* < 0.05). There were no significant difference in feces microbial Asp, Thr and His of AA composition in growing pigs between the 12% CP and 18% CP groups (*p* > 0.05).

The ileal microbial Cys and Met of AA composition was different among the three experimental groups (*p* < 0.01), i.e., greatest in the 18% CP group, greater in the 12% CP group and lowest in the 15% CP group (*p* < 0.05). The ileal microbial Leu of AA composition was different among the three experimental groups (*p* < 0.01), i.e., greater in the 15% CP and 18% CP groups than in the 12% CP group (*p* < 0.05). There were significant differences in feces microbial Leu of AA composition in growing pigs between the 18% CP group and the 15% CP group (*p* < 0.05). The ileal microbial Phe, EAA and EAA/NEAA of AA composition was different among the three experimental groups (*p* < 0.01), i.e., greatest in the 18% CP group and lowest in the 12% CP group (*p* < 0.05). The ileal microbial NEAA of AA composition was different among the three experimental groups (*p* < 0.01), i.e., greater in the 12% CP group than in the 15% CP and 18% CP groups (*p* < 0.05). There were no significant differences in feces microbial NEAA of AA composition in growing pigs between the 15% CP and 18% CP groups (*p* > 0.05). There were no significant differences in feces microbial Tyr and Lys of AA composition in growing pigs among the three experimental groups (*p* > 0.05). The ileal microbial Arg of AA composition was different among the three experimental groups (*p* < 0.01), i.e., greatest in the 15% CP group, greater in the 12% CP group and lowest in the 18% CP group (*p* < 0.05).

## 4. Discussion

### 4.1. Fecal N Excretion

Livestock and poultry excreta are important sources of environmental N pollution. In feeding and management, it has become an effective way to reduce nitrogen emissions by reducing the amount of protein raw materials to reduce dietary protein levels, and then by adding synthetic AA, to meet animal AA needs of low-protein diets [16]. Compared to feeding them the 18% CP diet, fecal N losses decreased by 7.4% and 18.3% when feeding the 15% and 12% (protein-free) CP diet, respectively. Our values are lower than those reported previously. Gloaguen et al. [17] pointed out that dietary protein levels decreased from 20% to 17%, with an average reduction of 1 percentage point, and total nitrogen emissions decreased by 8.04%. At the same time, with the decrease of dietary protein levels, N intake, N uptake and N deposition decreased, while N apparent utilization efficiency and biological potency increased. In addition, previous study has shown that N excretion was reduced by 10% and 40% when decreasing dietary CP levels from 15% to 6% in growing pigs [18]. In the present study, our greatest contribution to decreasing the fecal N losses when reducing CP concentration from 18% to 12% is protein N origin. We also found that feces microbial N of growing pigs had no obvious change among the 12%, 15% and 18% CP diets; therefore, microbial N in feces is not a principal factor to reduce total N-losses. However, other research suggests that Bacterial N, combined with ammonia and urea N, accounted for approximately 61% of total nitrogenous losses in pigs which were fed free protein diets or casein diets [13]. Dietary manipulation to reduce NH_3_ emission seems highly effective for realistic and efficient practice to reduce NH_3_ emission throughout the process from feeding the animal to spreading the fertilizer [19]. Dietary regulation to decrease NH_3_ emissions seems to be a highly effective method to cut back on NH_3_ emissions. We also know that reducing dietary CP levels is one effective way to decrease ammonia emitted from pigs houses in industry [20].

### 4.2. AA Digestibility

Added essential crystalline AA to low-protein diet can not only increase AA digestibility and reduce AA excretion, but also provide a better balance of AA absorption for the whole intestine by decreasing the competition of intestinal epithelial cells for AA transport [18]. The research showed that adding crystalline AA to diets could improve the digestible energy, metabolizable energy and net energy of growing pigs [21]. Similarly, adding crystalline AA to low-protein diets can improve performance and carcass traits of late-breeding sows [16]. Other studies have shown that adding branched chain AA to low-protein diets can improve the growth performance in pigs to a level similar to that of adequate protein diets [22]. In addition, crystalline AA can provide growth requirements for broilers in low-protein diets [23]. In conclusion, added EAA to low-protein diet can make up for the problem of protein deficiency, and even improve animal production performance and lipid metabolism.

### 4.3. Microbial AA Composition

The present study indicated that the effects of dietary N substrates on ileal microbial AA composition were more significant than in feces. Ileal microbial EAA and NEAA compositions were affected by dietary AA composition, while fecal microbial EAA and NEAA compositions were not affected by dietary AA composition. This is related to the different microbiome composition from anterior to posterior [24]. Up until now, very little research has addressed the microbial AA composition of pigs. Yet, the decomposition and synthesis of N and AA in intestinal microflora were very active. Previous studies have shown that NEAA added to a low-protein Thr-deficient diet can improve the utilization of these AAs for protein deposition in young pigs [25]. Branched chain AA, Lys and Phe, synthesized by swine intestinal bacteria with intestinal fistula, can be absorbed in the small intestine and large intestine [26,27,28]. In addition, prior research has suggested that a carbohydrate to nitrogenous compound ratio in the large intestine is an determinant factor of microbial metabolism and gut barrier function in the colon [29]. These findings will be beneficial to enhancing our understanding of the fact that microbe-related secondary bile acid metabolism may mediate the interplay between intestinal barrier functions and microbes.

## 5. Conclusions

In conclusion, we had direct implications for reducing dietary CP levels from 18% to 12%, including a decreased fecal N excretion, and a principal factor to reducing total N losses in feces was dietary N except for microbial N. Microbial EAA and NEAA composition in ileum was affected by the AA composition in growing pigs’ diets, while dietary AA composition had no effects on the composition of microbial EAA and NEAA in feces.

## Figures and Tables

**Table 1 animals-10-02092-t001:** Ingredients and nutrient contents of diets in 30–60 kg growing pigs (air-dry basis).

Items	18% CP	15% CP	12% CP	Nutrition Levels	18% CP	15% CP	12% CP
Corn	58.60	67.50	77.60	DE (MJ/kg)	14.20	14.20	14.20
Soy meal	29.00	19.50	10.00	CP, %	18.27	15.16	12.35
Wheat bran	7.80	6.94	5.06	Lys, %	0.97	0.97	0.94
Soy oil	1.55	2.38	3.00	Met + Cys, %	0.57	0.56	0.55
Monocalcium phosphate	0.69	0.78	0.90	Thr, %	0.61	0.61	0.60
Stone dust	0.87	0.89	0.90	Arg, %	1.08	0.82	0.57
NaCl	0.30	0.30	0.30	Trp, %	0.17	0.17	0.17
Vitamin-mineral Premix	1.00	1.00	1.00	His, %	0.41	0.33	0.25
L-Lysine HCl (Lys)	0.18	0.46	0.74	Phe, %	0.77	0.62	0.46
DL-Methionine (Met)	0.00	0.09	0.17	Val, %	0.66	0.56	0.44
L-Threonine (Thr)	0.01	0.14	0.26	Ile, %	0.64	0.49	0.35
L-Tryptophan (Trp)	0.00	0.02	0.07	Leu, %	1.35	1.14	0.94
Total	100.00	100.00	100.00	Available *p*, %	0.51	0.48	0.45
				EAA, %	7.00	6.04	5.03
				NEAA, %	9.12	7.50	5.83
				EAA/NEAA	0.76	0.81	0.86

**Table 2 animals-10-02092-t002:** Feces flow amount and apparent digestibility of crude protein (CP) and N of growing pigs fed low protein diets.

Items	18% CP	15% CP	12% CP	SEM	*p*-Values
Fecal CP and N flow amount (mg/g DM)					
CP	245.80 ^a^	228.60 ^b^	201.50 ^c^	2.70	<0.01
TN	39.40 ^a^	36.50 ^b^	32.20 ^c^	0.43	<0.01
NPN	4.80 ^a^	3.25 ^b^	3.20 ^b^	0.28	<0.01
PN	34.60 ^a^	33.30 ^a^	29.10 ^b^	0.50	<0.01
Total microbial N	5.68 ^c^	6.60 ^a^	5.99 b	0.07	<0.01
Microbial NPN	0.64	0.70	0.64	0.02	0.14
Microbial PN	5.05 ^c^	5.90 ^a^	5.35 ^b^	0.06	<0.01
Feces apparent digestibility (%)					
CP	85.30 ^c^	88.10 ^b^	91.00 ^a^	0.52	<0.01
DM	89.50 ^c^	92.00 ^b^	94.60 ^a^	0.41	<0.01
TN	85.30 ^c^	88.10 ^b^	91.00 ^a^	0.52	<0.01
NPN	83.10 ^c^	87.40 ^b^	90.60 ^a^	0.93	<0.01
PN	97.90 ^c^	98.30 ^b^	98.70 ^a^	0.77	<0.01

^a,b,c^ Values in the same row with different letter superscripts mean significant differences (*p* < 0.05).

**Table 3 animals-10-02092-t003:** Feces Total AA, essentials AA, nonessential AA flow amount (mg/g DM) and apparent digestibility (%) with or without microbia of growing pigs fed low protein diets.

Items	18% CP	15% CP	12% CP	SEM	*p*-Values
Feces with microbes					
NEAA flow amount	90.50 ^a^	73.90 ^b^	76.20 ^b^	1.46	<0.01
EAA flow amount	86.20 ^a^	75.00 ^b^	72.40 ^b^	1.44	<0.01
TAA flow amount	176.80 ^a^	148.80 ^b^	148.60 ^b^	2.57	<0.01
NEAA digestibility	92.50 ^b^	93.00 ^b^	94.60 ^a^	0.29	<0.01
EAA digestibility	90.70 ^b^	90.80 ^b^	92.90 ^a^	0.36	<0.01
TAA digestibility	91.70 ^b^	92.10 ^b^	93.90 ^a^	0.32	<0.01
EAA/NEAA flow amount	0.95 ^b^	1.01 ^a^	0.95 ^b^	0.02	0.03
Feces without microbes					
NEAA flow amount	73.50 ^a^	54.40 ^b^	58.50 ^b^	1.49	<0.01
EAA flow amount	71.70 ^a^	57.80 ^b^	56.70 ^b^	1.40	<0.01
TAA flow amount	145.20 ^a^	111.90 ^b^	115.20 ^b^	2.55	<0.01
NEAA digestibility	93.80 ^c^	94.70 ^b^	95.70 ^a^	0.24	<0.01
EAA digestibility	92.40 ^b^	93.12 ^b^	94.55 ^a^	0.31	<0.01
TAA digestibility	93.20 ^c^	94.00 ^b^	95.20 ^a^	0.27	<0.01
EAA/NEAA flow amount	0.98 ^b^	1.06 ^a^	0.97 ^b^	0.02	0.03

^a,b,c^ Values in the same row with different letter superscripts mean significant differences (*p* < 0.05).

**Table 4 animals-10-02092-t004:** Feces amino acids flow amount (mg/g DM) of growing pigs fed low protein diets.

Items	18% CP	15% CP	12% CP	SEM	*p*-Values
Ser	3.30 ^a^	2.80 ^b^	2.80 ^b^	0.11	<0.01
Glu	24.00 ^a^	20.40 ^b^	20.30 ^b^	0.62	<0.01
Ala	19.80 ^a^	15.00 ^c^	17.20 ^b^	0.51	<0.01
Gly	15.80 ^a^	12.80 ^b^	11.90 ^b^	0.42	<0.01
Pro	9.80	8.90	10.10	0.39	0.13
Tyr	5.90 ^a^	5.10 ^b^	4.90 ^b^	0.21	0.01
Asp	17.90 ^a^	14.00 ^b^	13.10 ^b^	0.42	<0.01
Thr	7.20 ^a^	5.60 ^b^	4.90 ^b^	0.26	<0.01
Val	12.60 ^a^	9.20 ^c^	10.30 ^b^	0.30	<0.01
Cys	2.60 ^a^	2.40 ^b^	2.50 ^b^	0.05	0.01
Met	3.00	2.90	2.70	0.12	0.18
Ile	10.50 ^a^	9.70 ^ab^	8.80 ^b^	0.31	<0.01
Leu	19.00 ^a^	16.90 ^b^	17.10 ^b^	0.34	<0.01
Phe	8.70 ^a^	7.70 ^b^	7.60 ^b^	0.27	0.02
Lys	8.20 ^a^	7.70 ^a^	6.60 ^b^	0.33	0.01
His	4.10	4.30	3.70	0.18	0.09
Arg	4.50 ^a^	3.60 ^b^	4.30 ^a^	0.13	<0.01

^a,b,c^ Values in the same row with different letter superscripts mean significant differences (*p* < 0.05).

**Table 5 animals-10-02092-t005:** Feces apparent amino acids digestibility (%) in growing pigs low protein diets.

Items	18% CP	15% CP	12% CP	SEM	*p*-Values
Ser	94.80 ^b^	94.40 ^b^	95.50 ^a^	0.19	<0.01
Glu	92.00 ^b^	92.40 ^b^	93.90 ^a^	0.38	<0.01
Gly	85.00 ^b^	86.80 ^b^	89.90 ^a^	0.59	<0.01
Ala	87.40 ^b^	89.10 ^a^	90.90 ^a^	0.60	<0.01
Pro	97.10 ^b^	96.90 ^b^	97.60 ^a^	0.16	0.01
Tyr	91.50 ^b^	91.30 ^b^	93.60 ^a^	0.34	<0.01
Asp	90.00 ^c^	91.50 ^b^	93.00 ^a^	0.39	<0.01
Thr	91.30 ^b^	92.40 ^b^	94.70 ^a^	0.43	<0.01
Cys	92.20 ^a^	90.50 ^b^	91.20 ^a^	0.32	<0.01
Val	88.10 ^b^	90.20 ^a^	90.90 ^a^	0.55	<0.01
Met	91.70 ^b^	89.00 ^c^	93.20 ^a^	0.44	<0.01
Ile	88.50 ^b^	88.10 ^b^	91.00 ^a^	0.50	<0.01
Leu	90.70 ^b^	91.05 ^b^	93.20 ^a^	0.41	<0.01
Phe	90.20 ^b^	89.50 ^b^	91.50 ^a^	0.48	0.03
Lys	91.50 ^b^	91.20 ^b^	94.60 ^a^	0.43	<0.01
His	92.40 ^a^	90.30 ^b^	93.30 ^a^	0.49	<0.01
Arg	96.70	97.10	97.10	0.18	0.21

^a,b,c^ Values in the same row with different letter superscripts mean significant differences (*p* < 0.05).

**Table 6 animals-10-02092-t006:** Feces microbial amino acids flow amount (mg/g DM) of growing pigs fed low protein diets.

Items	18% CP	15% CP	12% CP	SEM	*p*-Values
Ser	0.58 ^b^	0.84 ^a^	0.50 ^c^	0.02	0.08
Glu	3.99	4.49	4.20	0.14	<0.01
Gly	2.43 ^c^	2.83 ^a^	2.62 ^b^	0.06	<0.01
Ala	3.08 ^c^	3.81 ^a^	3.56 ^b^	0.07	<0.01
Pro	1.38 ^c^	2.11 ^a^	1.98 ^b^	0.04	<0.01
Tyr	1.23 ^b^	1.41 ^a^	1.20 ^b^	0.04	<0.01
Asp	4.33 ^a^	4.01 ^b^	3.67 ^c^	0.07	<0.01
Thr	1.30 ^c^	1.66 ^a^	1.46 ^b^	0.05	<0.01
Cys	0.38 ^a^	0.28 ^b^	0.26 ^b^	0.03	0.01
Val	1.74 ^b^	2.36 ^a^	1.81 ^b^	0.05	<0.01
Met	0.58 ^b^	0.69 ^a^	0.55 ^b^	0.02	0.02
Ile	1.38 ^c^	2.35 ^a^	2.03 ^b^	0.04	<0.01
Leu	3.59	3.74	3.79	0.07	0.12
Phe	1.51 ^b^	1.71 ^a^	1.54 ^b^	0.05	0.04
Lys	1.42 ^a^	1.50 ^a^	1.35 ^b^	0.04	0.05
His	0.90 ^a^	0.91 ^a^	0.78 ^b^	0.03	0.01
Arg	1.72 ^b^	2.19 ^a^	2.17 ^a^	0.06	<0.01
TAA	31.50 ^c^	36.90 ^a^	33.40 ^b^	0.36	<0.01
NEAA	17.00 ^b^	19.50 ^a^	17.70 ^b^	0.26	<0.01
EAA	14.50 ^c^	17.40 ^a^	15.70 ^b^	0.20	<0.01
EAA/NEAA	0.85	0.89	0.89	0.01	0.16

^a,b,c^ Values in the same row with different letter superscripts mean significant differences (*p* < 0.05).

**Table 7 animals-10-02092-t007:** Feces microbial amino acids composition (%) of growing pigs fed low protein diets.

Items	18% CP	15% CP	12% CP	SEM	*p*-Values
Ser	1.83 ^b^	2.28 ^a^	1.49 ^c^	0.06	<0.01
Glu	12.64	12.16	12.54	0.34	0.57
Gly	7.70	7.68	7.83	0.18	0.80
Ala	9.8 ^b^	10.3 ^a^	10.70 ^a^	0.21	0.03
Pro	4.39 ^b^	5.71 ^a^	5.89 ^a^	0.12	<0.01
Tyr	3.91	3.83	3.59	0.12	0.11
Asp	13.7 ^a^	10.9 ^b^	11.0 ^b^	0.19	<0.01
Thr	4.13	4.51	4.36	0.12	0.12
Cys	1.20 ^a^	0.77 ^b^	0.76 ^b^	0.08	<0.01
Val	5.53 ^b^	6.39 ^a^	5.44 ^b^	0.14	<0.01
Met	1.85	1.86	1.64	0.06	0.04
Ile	4.37 ^b^	6.38 ^a^	6.06 ^a^	0.13	<0.01
Leu	11.4 ^a^	10.1 ^b^	11.30 ^a^	0.23	<0.01
Phe	4.78	4.63	4.61	0.15	0.66
Lys	4.50 ^a^	4.08 ^b^	4.04 ^b^	0.12	0.03
His	2.83 ^a^	2.47 ^b^	2.32 ^b^	0.09	<0.01
Arg	5.44 ^c^	5.93 ^b^	6.45 ^a^	0.16	<0.01
TAA	100.00	100.00	100.00	0.00	1.00
NEAA	54.00	52.80	53.00	0.42	0.16
EAA	46.00	47.10	47.00	0.42	0.16
EAA/NEAA	0.85	0.89	0.89	0.01	0.16

^a,b,c^ Values in the same row with different letter superscripts mean significant differences (*p* < 0.05).

**Table 8 animals-10-02092-t008:** Ileal microbial amino acids composition (%) of growing pigs fed low protein diets.

Items	18% CP	15% CP	12% CP	SEM	*p*-Values
Ser	3.23 ^b^	4.03 ^a^	4.36 ^a^	0.12	<0.01
Glu	12.84 ^a^	9.74 ^b^	9.04 ^b^	0.25	<0.01
Gly	4.93 ^c^	6.54 ^a^	5.59 ^b^	0.12	<0.01
Ala	6.00 ^c^	8.28 ^b^	8.96 ^a^	0.18	<0.01
Pro	4.85 ^b^	6.80 ^a^	6.30 ^a^	0.19	<0.01
Tyr	3.50	3.97	3.64	0.19	0.24
Asp	11.8 ^a^	8.8 ^b^	11.8 ^a^	0.23	<0.01
Thr	3.26 ^a^	2.26 ^b^	3.01 ^a^	0.12	<0.01
Cys	2.57 ^a^	1.36 ^c^	1.57 ^b^	0.05	<0.01
Val	5.31 ^b^	6.94 ^a^	6.88 ^a^	0.17	<0.01
Met	3.95 ^a^	2.24 ^c^	3.14 ^b^	0.15	<0.01
Ile	5.29 ^b^	6.80 ^a^	6.75 ^a^	0.28	<0.01
Leu	11.8 ^a^	11.8 ^a^	10.6 ^b^	0.23	<0.01
Phe	7.75 ^a^	6.91 ^b^	5.77 ^c^	0.12	<0.01
Lys	4.30	4.80	4.67	0.32	0.52
His	4.38 ^a^	3.71 ^b^	4.61 ^a^	0.19	0.01
Arg	4.22 ^b^	5.00 ^a^	3.26 ^c^	0.15	<0.01
TAA	100.00	100.00	100.00	0.00	1.00
NEAA	43.70 ^b^	44.20 ^b^	46.10 ^a^	0.23	<0.01
EAA	52.80 ^a^	51.80 ^b^	50.30 ^c^	0.26	<0.01
EAA/NEAA	1.21 ^a^	1.17 ^b^	1.09 ^c^	0.01	<0.01

^a,b,c^ Values in the same row with different letter superscripts mean significant differences (*p* < 0.05).

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
