# Peer review of "Effects of Dietary Crude Protein Levels on Fecal Crude Protein, Amino Acids Flow Amount, Fecal and Ileal Microbial Amino Acids Composition and Amino Acid Digestibility in Growing Pigs"

_animals, 2020, doi:10.3390/ani10112092_

Round 1

Reviewer 1 Report

The aim of this study was to determine the effects of different dietary crude protein levels on fecal crude protein content and amino acid flow, fecal crude protein and amino acids digestibility, fecal and ileal microbiota AA composition in growing pigs. The obtained research results provided important information for pig producers, pig compound feed plants, public opinion. The number of piss used in the experiment is sufficient, the test methods used are correct. The discussion is well carried out and exhausting. References well chosen. Before publishing in Animals, the paper requires additions and corrections. The list of proposed changes is given below:

General comments:

Please prepare the article according to the instructions for the authors.

  • The affiliation does not include the e-mail of each co-author of the article, the co-authors' initials (preferably the first letter of the first name and the first letter of the surname), for the corresponding author, please provide e-mail, phone number, initials
  • There are no chapters in the article: Simple Summary; Authors Contributions, Conflict of Interest
  • References section must follow the instructions for authors, no printed names, "[J] before the journal name, Abbreviated name journal, year of publication in bold, page range" - "with insert instead of keyboard, eg Author 1, Author 2 ,. Title of the article. Abbreviated Journal Name Year, Volume, page range.
  • Reference numbers in text without superscript [1] instead of [1]

Detailed comments:

L34 “Thr” or “Met”? (see Table 7)

L52 [1] instead of [1], this way References applies to the remaining citations

L53 "feed additives" instead of additives

L61 “(EAA)” instead of (Essential amino acids, EAA)

L63 "CP content reduction was more than 3%" instead of the current form, or no compensatory growth in pigs?

L66 [9] instead of (NRC 2012)

L69 mTOR and S6K1 full names in parentheses for the first time in the text

L75 add references after "... in production"

L96 reference number [ ] instead of NRC (2012)

L95 provide information about the lighting program (length, intensity, type)

L123 [15] instead of (SAS Institute ...)

L131 nonprotein instead of Non protein, lowercase and combined

L132 For N proteins only L-CP is significantly smaller than H-CP

L135,138 "fecals" in small letters

L136 nonprotein instead of non protein

L133 percentage of protein N flow in total N flow? Where is this?

L143-144 The CP, total N …… delete at the end of the description

L147 nonessential (NEAA) and essentials (EAA) amino acids flow and apparent total amino acid (TAA)…. instead of current form

L149 (P < 0.01) instead of (P < 0.001)

L151 "dot" after fecal?

L161 total AA (TAA), Essentials AA (EAA), nonessential AA (NEAA) flow ... instead of current form

L166 for "Ala" H-CP also significantly higher than L-CP and M-CP

L168 for Ala M-CP and L-CP are significant differences

L170 for "Leu" H-CP is higher than L-CP and M-CP (P < 0.05)

L172 inconsistency of the description with the data for "Leu" M-CP and H-CP the differences are significant

L186 “Met” also? L-CP higher than M-CP and H-CP

L191-192 Repeating the description from L186-187

L201 description not compatible with the data "While there were no, .., for Ala and Val, the description was made in L197

L204-207 description not compatible with data for Ala and Val, the description was made in L197

L211 "Val and" instead of Valand

L212 groups (P < 0.05) instead of current form

L221 between group and (P < 0.05) space

L228 "Leu and EAA/NEEA" instead of current form

L233 no significant differences also between L-CP and M-CP for "Ala, Pro, Ile, His"

In table 7 EAA/NEAA instead of E/NE

 L230 "Thr and" instead of current form

L273-275 repetition, see L270-271

L282 for "Leu" M-CP and H-CP higher than L-CP, description inconsistent with Table 8

L283 the difference between M-CP and H-CP is significant

 Notes on other chapters are provided in the general commentary

Author Response

animals-964715 modify description

Dear Professor,

I would like to thank the editor for reviewing this article in the midst of his busy schedule. Your comprehensive guidance to our work is our way forward. We are very grateful to the external audit experts for their expert opinions, which not only make us clearly aware of our shortcomings, but also is great significant for our future study and work. We have seriously revised the questions raised by the experts and teachers, and if there are any omissions, we will carefully check the amendments again. Finally, I sincerely appreciate all the teachers who have devoted their time to this article. Finally, I offer my sincerest thank you again.

The following are our revisions of this article, please check again.

Question 1: L34 “Thr” or “Met”? (see Table 7)

Answer 1: Owing to your suggestion, we looked at the table and replaced "Met" with "Thr".

Question 2: L52 [1] instead of [1], this way References applies to the remaining citations

Answer 2: Owing to your suggestion, we revised the way all the references in the article were written.

Question 3: L53 "feed additives" instead of additives

Answer 3: Owing to your suggestion, we replaced "additives" with "feed additives".

Question 4: L61 “(EAA)” instead of (Essential amino acids, EAA)

Answer 4: Owing to your suggestion, we will replace "Essential amino acids, EAA" with "EAA".

Question 5: L63 "CP content reduction was more than 3%" instead of the current form, or no compensatory growth in pigs?

Answer 5: we agree and have use "CP content reduction was more than 3%" instead of the current form.

Question 6: L66 [9] instead of (NRC 2012)

Answer 6: Owing to your suggestion, we replaced "(NRC 2012)" with "[9]".

Question 7: L69 mTOR and S6K1 full names in parentheses for the first time in the text

Answer 7: Owing to your suggestion, we rewrote mTOR and S6K1 to "mammalian target of rapamycin (mTOR)" and "S6 kinase 1 (S6K1)".

Question 8: L75 add references after "... in production"

Answer 8: Owing to your suggestion, we looked up the relevant references, as follows:

Russell H. Chittenden. Discussion On The Merits Of A Relatively Low Protein Diet. 1911, 2(2647):656-667.

Question 9: L96 reference number [ ] instead of NRC (2012)

Answer 9: Owing to your suggestion, we replaced "NRC (2012)" with [9].

Question 10: L95 provide information about the lighting program (length, intensity, type)

Answer 10: Owing to your suggestion, we supplement the lighting information of experimental animals in this paper.

Question 11: L123 [15] instead of (SAS Institute ...)

Answer 11: Owing to your suggestion, we supplement the relevant references instead of "SAS Institute."

Question 12: L131 nonprotein instead of Non protein, lowercase and combined

Answer 12: Owing to your suggestion, we replaced "Non protein" with "nonprotein".

Question 13: L132 For N proteins only L-CP is significantly smaller than H-CP

Answer 13: Owing to your suggestion, we deleted the wrong expression.

Question 14: L135,138 "fecals" in small letters

Answer 14: Owing to your suggestion, we corrected the way the words were written.

Question 15: L136 nonprotein instead of non protein

Answer 15: Owing to your suggestion, we replaced "non protein" with "nonprotein".

Question 16: L133 percentage of protein N flow in total N flow? Where is this?

Answer 16: Owing to your suggestion, we have rewritten the whole paragraph as follows:

Fecal N, CP flow and digestibility, as affected by dietary CP concentration are presented in Table 2. Fecal apparent DM, CP, total N, protein N and nonprotein N digestibility and the CP, total N flow amount were different among 3 treatments (P < 0.01), i.e. greatest in the L-CP group, lowest in the H-CP group (P < 0.05). The protein N flow amount of feces in the L-CP groups were lower than that in the M-CP and H-CP group (P < 0.05). There are no significant differences in Feces protein N flow of pigs between the M-CP group and the H-CP group (P > 0.05). The Nonprotein N flow amount of feces in the L-CP and M-CP groups were lower than that in the H-CP group (P < 0.05). There are no significant differences in feces protein N flow of pigs between M-CP group and L-CP groups (P > 0.05). Total microbial N, microbial protein N of feces were different among 3 treatments (P < 0.01), i.e. greatest in the M-CP group, greater in the L-CP group, lowest in the H-CP group (P < 0.05). However, there were no significant differences in fecal microbial non-protein N among 3 treatments (P = 0.14).

Question 17: L143-144 The CP, total N …… delete at the end of the description

Answer 17: Owing to your suggestion, we deleted the unnecessary description.

Question 18: L147 nonessential (NEAA) and essentials (EAA) amino acids flow and apparent total amino acid (TAA)…. instead of current form

Answer 18: Owing to your suggestion, we use acronyms instead of the original expressions.

Question 19: L149 (P < 0.01) instead of (P < 0.001)

Answer 19: Owing to your suggestion, we replaced "(P < 0.001)" with "(P < 0. 01)".

Question 20: L151 "dot" after fecal?

Answer 20: Owing to your suggestion, we added the correct punctuation after "fecal".

Question 21: L161 total AA (TAA), Essentials AA (EAA), nonessential AA (NEAA) flow ... instead of current form

Answer 21: Owing to your suggestion, we use acronyms instead of the original expressions.

Question 22: L166 for "Ala" H-CP also significantly higher than L-CP and M-CP

Answer 22: Owing to your suggestion, we checked the data in the table and added "Ala" to the sentence.

Question 23: L168 for Ala M-CP and L-CP are significant differences

Answer 23: Owing to your suggestion, we checked the data in the table and modified the wrong expression.

Question 24: L170 for "Leu" H-CP is higher than L-CP and M-CP (P < 0.05)

Answer 24: Owing to your suggestion, we checked the data in the table and modified the wrong expression.

Question 25: L172 inconsistency of the description with the data for "Leu" M-CP and H-CP the differences are significant

Answer 25: Owing to your suggestion, we checked the data in the table and modified the wrong expression.

Question 26: L186 “Met” also? L-CP higher than M-CP and H-CP

Answer 26: Owing to your suggestion,

Question 27: L191-192 Repeating the description from L186-187

Answer 27: Owing to your suggestion, we deleted the repeated sentences.

Question 28: L201 description not compatible with the data "While there were no, .., for Ala and Val, the description was made in L197

Answer 28: Owing to your suggestion, we checked the data in the table and modified the wrong expression.

Question 29: L204-207 description not compatible with data for Ala and Val, the description was made in L197

Answer 29: Owing to your suggestion, we checked the data in the table and modified the wrong expression.

Question 30: L211 "Val and" instead of Valand

Answer 30: Owing to your suggestion, we added a space between "Valand".

Question 31: L212 groups (P < 0.05) instead of current form

Answer 31: Owing to your suggestion, we added a space after "groups"

Question 32: L221 between group and (P < 0.05) space

Answer 32: Owing to your suggestion, we added a space after "groups"

Question 33: L228 "Leu and EAA/NEEA" instead of current form

Answer 33: Owing to your suggestion, we replaced "E/NE" with "EAA/NEEA".

Question 34: L233 no significant differences also between L-CP and M-CP for "Ala, Pro, Ile, His"

Answer 34: Owing to your suggestion, we checked the data and added "Ala, Pro, Ile, His" to the whole sentence.

Question 35: In table 7 EAA/NEAA instead of E/NE

Answer 35: Owing to your suggestion, we replaced "E/NE" with "EAA/NEEA".

Question 36:  L230 "Thr and" instead of current form

Answer 36: Owing to your suggestion, we added a space between "Thrand".

Question 37: L273-275 repetition, see L270-271

Answer 37: Owing to your suggestion, we checked and deleted the duplicates.

Question 38: L282 for "Leu" M-CP and H-CP higher than L-CP, description inconsistent with Table 8

Answer 38: Owing to your suggestion, we checked and modified the wrong description.

Question 39: L283 the difference between M-CP and H-CP is significant

Answer 39: Owing to your suggestion, we checked and modified the wrong expression.

Question 40: Notes on other chapters are provided in the general commentary

Answer 40: We have not only corrected the above errors, but also made comprehensive corrections to other errors in the article. All the changes are traced in the text.

Reviewer 2 Report

authors must carefully review the paper.

Simple summary

Line 17: please delete “fed”

Abstract

Line 25: please delete “fed”

Line 32: …“while the flow velocity of ….” Please specify which group

Line 33: In which table is reported the AA means values of M-CP?

Line 34: “Fecal TTA and NEAA….also increased linearly”  looking at the table data linearly increases only fecal TTA.

Introduction

Line 61: please check the relevance of citation

Line 74: change “….index. therefore….” In ”…index. Therefore…”

Line 132-133: in table 2 the protein N flow in M-PC and H-PC is the same for statistical analysis

Line 134-135: the authors' claim is unclear. The authors are asked to clarify, because if the data are those reported in table 2 (second part) the unit of measurement is mg/g DM and not a percentage.

Line 139: please change L-CP group in H-CP group.

Line 141: please delete “…greater in the L-CP group. Therefore, the statement changes in “i.e. greatest in M-CP group and lowest in the H-CP group …”

Line 152: “…M-CP group. feaces…” change in “…M-CP group. Feaces…” Please the authors make uniform “feaces” in the paper.

Line 169: Please the authors delete Ala and change in Cys

Line 173: Please the authors delete Leu. No significant difference is only for Lys in M-CP and H-CP groups.

Line 177 “…betweenthe…” please change in “…between the…”

Line 205: please change “…i.e. greatest in the L-CP group, greater in the H-CP group and…” in “… greatest in L-CP group and….”

Line 206: the statement reported in line206-207 “Feces Ala…… (P<0.05).” is related to table 3 or 4? Please the authors verify the claim.

Line 212: change “…Valand…” in “…Val and…”

Line 229: change “…Leuand…” in “…Leu and…”

Line 244 change “…thatin…” in “…that in…”

Line 261: change “…Thrand…” in “…Thr and…”

Line 274-276: please the authors delete the statement because is just written in line 271-272.

Line 276: please the authors to verify the statement “….i.e. greatest in the M-CP and lowest in H-CP” because is true only for ileal microbial Gly

Line 284: please change L-CP group in H-CP group.

Line 288-290: please the authors delete the statement because is just written in line 283-285.

Line 299: “….Glougen et al…..” change in “….Glougen et al. [16]….

Line 303: change “In addation…” in “ In addition…”

Line 320: the authors Mao , Guo et al., 2020 carried out the test with gilts and not sows.

Table 7: E/NE please change in EAA/NEAA

The bibliography does not attend the guideline

Author Response

animals-964715 modify description

Dear Professor,

I would like to thank you for reviewing this article in the midst of his busy schedule. Your comprehensive guidance to our work is our way forward. We are very grateful to the external audit experts for their expert opinions, which not only make us clearly aware of our shortcomings, but also is great significant for our future study and work. We have seriously revised the questions raised by the experts and teachers, and if there are any omissions, we will carefully check the amendments again. Finally, I sincerely appreciate all the teachers who have devoted their time to this article. Finally, I offer my sincerest thank you again.

The following are our revisions of this article, please check again.

Question 1: Line 17: please delete “fed”

Answer 1: Owing to your suggestion, we deleted the word "fed".

Abstract

Question 2: Line 25: please delete “fed”

Answer 2: Owing to your suggestion, we deleted the word "fed".

Question 3: Line 32: …“while the flow velocity of ….” Please specify which group

Question 4: Line 33: In which table is reported the AA means values of M-CP?

Question 5: Line 34: “Fecal TTA and NEAA….also increased linearly”  looking at the table data linearly increases only fecal TTA.

Answer 3,4, 5: We have rewrite this part as following: Compared with M-CP group and L-CP group, the flow amount of Asp, Ser, Glu, Gly, Tyr, Val, Ile, Leu and Phe in feces of piglets in H-CP group increased significantly, while the flow amount of Arg in M-CP group was lower than that in L-CP group and H-CP group, while the fecal microbial N and AA of M-CP group were higher than those of the other two groups. Fecal TAA and NEAA flow amount increased linearly with the increase of dietary crude protein level from 12% to 18%.

Introduction

Question 6: Line 61: please check the relevance of citation

Answer 6: Owing to your suggestion, we reviewed and revised the references.

Question 7: Line 74: change “….index. therefore….” In ”…index. Therefore…”

Answer 7: Owing to your suggestion, we corrected the wrong words.

Question 8: Line 132-133: in table 2 the protein N flow in M-PC and H-PC is the same for statistical analysis

Question 9: Line 134-135: the authors' claim is unclear. The authors are asked to clarify, because if the data are those reported in table 2 (second part) the unit of measurement is mg/g DM and not a percentage.

Answer 8 and 9: We revised this paragraph: Fecal nitrogen, CP flow and digestibility, as affected by dietary CP concentration are presented in Table 2. Fecal apparent DM, CP, total N, protein N and nonprotein N digestibility and the CP, total N flow amount were different among 3 treatments (P < 0.01), i.e. greatest in the L-CP group, lowest in the H-CP group (P < 0.05). The protein N flow amount of feces in the L-CP groups were lower than that in the M-CP and H-CP group (P < 0.05). There are no significant differences in Feces protein N flow of pigs between the M-CP group and the H-CP group (P > 0.05). The Nonprotein N flow amount of feces in the L-CP and M-CP groups were lower than that in the H-CP group (P < 0.05). There are no significant differences in feces protein N flow of pigs between M-CP group and L-CP groups (P > 0.05). Total microbial N, microbial protein N of feces were different among 3 treatments (P < 0.01), i.e. greatest in the M-CP group, greater in the L-CP group, lowest in the H-CP group (P < 0.05). However, there were no significant differences in fecal microbial non-protein N among 3 treatments (P = 0.14).

Question 10: Line 139: please change L-CP group in H-CP group.

Answer 10: Owing to your suggestion, we changed "L-CP" to "H-CP".

Question 11: Line 141: please delete “…greater in the L-CP group. Therefore, the statement changes in “i.e. greatest in M-CP group and lowest in the H-CP group …”

Answer 11: Owing to your suggestion, we deleted “…greater in the L-CP group, and added "and".

Question 12: Line 152: “…M-CP group. feaces…” change in “…M-CP group. Feaces…” Please the authors make uniform “feaces” in the paper.

Answer 12: Owing to your suggestion, we add appropriate punctuation after “…M-CP group”.

Question13: Line 169: Please the authors delete Ala and change in Cys

Answer 13: Owing to your suggestion, we changed "Ala" to "Cys"

Question14: Line 173: Please the authors delete Leu. No significant difference is only for Lys in M-CP and H-CP groups.

Answer 14: Owing to your suggestion, we deleted the word "Leu" from the sentence.

Question 15: Line 177 “…betweenthe…” please change in “…between the…”

Answer 15: Owing to your suggestion, we added a space between "betweenthe".

Question 16: Line 205: please change “…i.e. greatest in the L-CP group, greater in the H-CP group and…” in “… greatest in L-CP group and….”

Answer 16: Owing to your suggestion, we checked and deleted the extra description.

Question 17: Line 206: the statement reported in line206-207 “Feces Ala…… (P<0.05).” is related to table 3 or 4? Please the authors verify the claim.

Answer 17: we have corrected as: Feces Ala and Val flow amount in the H-CP group were greater than that in the M-CP and L-CP groups (P < 0.05).

Question 18: Line 212: change “…Valand…” in “…Val and…”

Answer 18: Owing to your suggestion, we added a space in the middle of "Valand".

Question 19: Line 229: change “…Leuand…” in “…Leu and…”

Answer 19: Owing to your suggestion, we added a space in the middle of "Leuand".

Question20: Line 244 change “…thatin…” in “…that in…”

Answer 20: Owing to your suggestion, we added a space in the middle of "thatin".

Question 21: Line 261: change “…Thrand…” in “…Thr and…”

Answer 21: Owing to your suggestion, we added a space in the middle of "Thrand".

Question 22: Line 274-276: please the authors delete the statement because is just written in line 271-272.

Answer 22: Owing to your suggestion, we checked and deleted the duplicates.

Question 23: Line 276: please the authors to verify the statement “….i.e. greatest in the M-CP and lowest in H-CP” because is true only for ileal microbial Gly

Answer 23: After careful examination, we deleted the word "and Arg", leaving only "The ileal microbial Gly of AA composition were different among 3 treatments (P < 0. 01), greatest in the M-CP group and lowest in the H-CP group (P < 0. 05)."

Question 24: Line 284: please change L-CP group in H-CP group.

Answer 24: Owing to your suggestion, we changed "L-CP" to "H-CP".

Question 25: Line 288-290: please the authors delete the statement because is just written in line 283-285.

Answer 25: Owing to your suggestion, we deleted the repeated parts of the sentence.

Question 26: Line 299: “….Glougen et al…..” change in “….Glougen et al. [16]….

Answer 26: Owing to your suggestion, we corrected the position of the references.

Question 27: Line 303: change “In addation…” in “ In addition…”

Answer 27: Owing to your suggestion, we modified the wrong words.

Question28: Line 320: the authors Mao , Guo et al., 2020 carried out the test with gilts and not sows.

Answer 28: Owing to your suggestion, we replaced the correct references.

Question 29: Table 7: E/NE please change in EAA/NEAA

Answer 29: Owing to your suggestion, we replaced "E/NE" with "EAA/NEAA".

Question 30:The bibliography does not attend the guideline

Answer 30: Owing to your suggestion, we revised all the references.

Reviewer 3 Report

The manuscript "Effects of dietary crude protein levels on fecal crude protein, amino acid flow, amino acid digestibility, fecal and ileal microbial ecology and amino acid composition in growing pigs" shows the effects of a reducing dietary CP concentration on the efficiency of protein utilization and reduction of nitrogen emissions. The results obtained are really interesting and well explained.

Only some minor points:

page 1 (Simple summary and Abstract) lines 17 and 25 "fed fed" correct

page 5 line 132 "Nonprotein N" please replaces the first uppercase "N" with the lowercase letter

page 7 line 177 space "betweenthe"

Please explain the extension of SEM, this acronym is introduced without an explanation

Please control the "References": they are written without following the instructions given in the "Guide for authors": in particular check the punctuation between the names of the authors, remove the "J" between the square brackets, write the name of the journal abbreviated and in italics.

Author Response

animals-964715 modify description

Dear Professor,

I would like to thank the editor for reviewing this article in the midst of his busy schedule. Your comprehensive guidance to our work is our way forward. We are very grateful to the external audit experts for their expert opinions, which not only make us clearly aware of our shortcomings, but also is great significant for our future study and work. We have seriously revised the questions raised by the experts and teachers, and if there are any omissions, we will carefully check the amendments again. Finally, I sincerely appreciate all the teachers who have devoted their time to this article. Finally, I offer my sincerest thank you again.

The following are our revisions of this article, please check again.

The manuscript "Effects of dietary crude protein levels on fecal crude protein, amino acid flow, amino acid digestibility, fecal and ileal microbial ecology and amino acid composition in growing pigs" shows the effects of a reducing dietary CP concentration on the efficiency of protein utilization and reduction of nitrogen emissions. The results obtained are really interesting and well explained.

Only some minor points:

Question 1: page 1 (Simple summary and Abstract) lines 17 and 25 "fed fed" correct

Answer 1: Owing to your suggestion, we deleted the repeated words in the sentence.

Question 2: page 5 line 132 "Nonprotein N" please replaces the first uppercase "N" with the lowercase letter

Answer 2: Owing to your suggestion, we replaced "Nonprotein N" with "nonprotein N".

Question 3: page 7 line 177 space "betweenthe"

Answer 3: Owing to your suggestion, we added a space between "betweenthe".

Question 4: Please explain the extension of SEM, this acronym is introduced without an explanation

Answer 4: Owing to your suggestion, we replaced "SEM" with "standard error of mean (SEM)".

Question5: Please control the "References": they are written without following the instructions given in the "Guide for authors": in particular check the punctuation between the names of the authors, remove the "J" between the square brackets, write the name of the journal abbreviated and in italics.

Question5: Please control the "References": they are written without following the instructions given in the "Guide for authors": in particular check the punctuation between the names of the authors, remove the "J" between the square brackets, write the name of the journal abbreviated and in italics.

Answer 5: Owing to your suggestion, we modified all the reference formats as required.

Reviewer 4 Report

Please proofread the manuscript. THERE ARE SEVERAL typos and grammatical errors

Ln 102: What was the rationale for selecting these time points for feces collection?

Ln 110: Whether the feces from all 5 days from each animal were pooled or daily analysis was performed?

Ln 112: How did the authors confirm the precipitate obtained at differential centrifugation was from microbial cells?

Ln 126: Expand SEM & SNK. Whether posthoc comparisons were performed?

Why a control group was not included with the standard CP ration?

Author Response

animals-964715 modify description

Dear Professor,

I would like to thank the editor for reviewing this article in the midst of his busy schedule. Your comprehensive guidance to our work is our way forward. We are very grateful to the external audit experts for their expert opinions, which not only make us clearly aware of our shortcomings, but also is great significant for our future study and work. We have seriously revised the questions raised by the experts and teachers, and if there are any omissions, we will carefully check the amendments again. Finally, I sincerely appreciate all the teachers who have devoted their time to this article. Finally, I offer my sincerest thank you again.

The following are our revisions of this article, please check again.

Question 1:Ln 102: What was the rationale for selecting these time points for feces collection?

Answer 1: Thank you for the question. Our aim was to explain whether 12% CP/15%CP/ 18% CP with the same Lys\Met+Cys\Thr\Trp level by the addition of four crystalline essential amino acids to affected the digestibility of protein-bound NEAA and EAA and the composition of microbial AA in ileum and feces for 30-d experiments, so we collected the feces on the d 25 to d 30.

Question 2:Ln 110: Whether the feces from all 5 days from each animal were pooled or daily analysis was performed?

Answer 2: Thank you for the question. The feces from all 5 days from each pigs were pooled in our study.

Question 3:Ln 112: How did the authors confirm the precipitate obtained at differential centrifugation was from microbial cells?

Answer 3: Thank you for the question. Materials with different specific gravity will be precipitated under different centrifugal force. Undigested food particles will be precipitated in 250 RCF 15 minutes and Micro cells exist in the supernatant. The supernatant include microbial cells can be precipitated at 14500 RCF for 30 min at 4 ℃. This method is based on the following literature.

METGES C C, PETZKE K J, EL-KHOURY A E, et al. Incorporation of urea and ammonia nitrogen into ileal and fecal microbial proteins and plasma free amino acids in normal men and ileostomates [J]. American Journal of Clinical Nutrition, 1999, 70(6): 1046-1058.

MINER-WILLIAMS W, MOUGHAN P J, FULLER M F. Endogenous Components of Digesta Protein from the Terminal Ileum of Pigs Fed a Casein-Based Diet [J]. Journal of Agricultural & Food Chemistry, 2009, 57(5): 2072-2078.

Question 4Ln 126: Expand SEM & SNK. Whether posthoc comparisons were performed?

Answer 4: We expanded standard error of mean (SEM) and the Student–Newman–Keuls (SNK) test in revised paper. We did the posthoc comparisons.

Question 5Why a control group was not included with the standard CP ration?

Answer 5: Our aim was to explain whether 12%CP/15%CP/ 18% CP(the standard CP ration) with the same Lys\Met+Cys\Thr\Trp level by the addition of four crystalline essential amino acids to affected the digestibility of protein-bound NEAA and EAA and the composition of microbial AA in ileum and feces. 18% CP is the standard CP ration in 30-60kg growing pigs. We will add this information in the revised manuscripts.

Round 2

Reviewer 1 Report

The aim of this study was to determine the effects of different dietary crude protein levels on fecal crude protein content and amino acid flow, fecal crude protein and amino acids digestibility, fecal and ileal microbiota AA composition in growing pigs. The obtained research results provided important information for pig producers, pig compound feed plants, public opinion. The number of piss used in the experiment is sufficient, the test methods used are correct. The discussion is well carried out and exhausting. References well chosen. Before publishing in Animals, the paper requires additions and corrections. The list of proposed changes is given below:

General comments:

The affiliation does not include the e-mail of each co-author of the article, the co-authors' initials (preferably the first letter of the first name and the first letter of the surname)

In References section page range" - "with insert instead of keyboard,

Detailed comments:

L30 Ileu also?

L33 NEAA flows also?

L34 Met too?

L64 is justification please

L75 adds references after "... in production"

L98 “[9]” - is correct?

In table 1, put a space between L-Lysine HCL and (Lys), DL-Methionine and (Met). Threonine a (Thr), L-Tryptopan a (Trp)

L111 without spaces "(250.,"

L131 N (PN) space

L132 N (TN) space

L133 N (NPN) space

L146 non-essential (NEAA) and essential (EAA) amino acids flow and apparent total amino acid number (TAA)…. instead of the current form

L151 "digestibility" instead of diestibility

L156 there are important differences between the M-CP group and the L-CP group, see Table 3

L161 Total AA (TAA), essentials AA (EAA), nonessential AA flow (NEAA) ... instead of actual form

L166 for "Ala" H-CP also significantly higher than L-CP and M-CP?

L168 for Ala M-CP and L-CP are significant differences

L170 for "Leu" H-CP is higher than L-CP and M-CP (P < 0.05)

In tables 4 to 8, add line before dates (the end of the header)

L180 M-CP instead of H-CP

L198-202 what is it?

L220 "groups H-CP than in groups M-CP and L-CP ...".

L233 -155 justification, no girdles and widows at the end of the line

There is no SEM value for the EAA / NEAA feature in Table 7

L266-268 repeating the description from lines 259-260

L271 "Ileu" instead of Leu

L273 the lowest for "Arg" was the L-CP group, not the H-CP group

L280 M-CP not H-CP

L281 "It was ....” please delete - repetition

In the references, the short name of the journal, not a printed character, must usually be letters

References 2-4, 7-9, 14, 16-20, 22, 25-26, 29 include all the authors of the articles, not just the first three.

Author Response

Editors

Animals

Nov 2, 2020

Re: check of “animals-964715”

"Effects of dietary different crude protein levels on fecal crude protein, amino acids flow, amino acid digestibility, fecal and ileal amino acids composition  and microbial ecology in growing pigs ".

Dear Editor:

I would like to thank the editor for reviewing this article in the midst of his busy schedule. Your comprehensive guidance to our work is our way forward. We are very grateful to the external audit experts for their expert opinions, which not only make us clearly aware of our shortcomings, but also is great significant for our future study and work. We have seriously revised the questions raised by the experts and teachers, and if there are any omissions, we will carefully check the amendments again. Finally, I sincerely appreciate all the teachers who have devoted their time to this article. Finally, I offer my sincerest thank you again.

Sincerely yours,

Zhiru Tang Ph.D, Professor,

Zhenguo Yang Ph.D,

Additional notes

In order to express the purpose of this experiment clear, We changed the presentation of the 3 experimental groups: replaced " L-CP" with "12% CP", replaced " M-CP" with "15% CP" and replaced "H-CP" with "18% CP".

In addition, we also corrected some errors we found ourselves and highlighted them in red.

Section : "Responses to the Comments by reviewer 1"

Comment 1: L30 Ileu also?

Authors'responses and locations of the revisions: Thanks. Owing to your suggestion, we carefully checked the experimental data and deleted Ile (Please see line 32 in the revised version ).

Comment 2: L33 NEAA flows also?

Authors'responses and locations of the revisions: Thanks. Owing to your suggestion, we carefully checked the experimental data and deleted NEAA.

The modified sentence as following:

Fecal TAA flow amount increased linearly with the increase of dietary crude protein level from 12% to 18%. (Please see line 36 in the revised version)

Comment 3: L34 Met too?

Authors'responses and locations of the revisions: Thanks. Owing to your suggestion, we examined it carefully and rewrote the sentence as "Except for Glu, Gly, Met, Tyr, Thr and Phe, there were significant differences among the three groups in the composition of 17 kinds of AAs in fecal microorganisms".  (Please see line 37-38 in the revised version)

Comment 4: L64 is justification please

Authors'responses and locations of the revisions: Thanks. Owing to your suggestion, we have corrected the format problems in the article.  (Please see line 64-66 in the revised version)

Comment 5: L75 adds references after "... in production"

Authors'responses and locations of the revisions: Thanks. Owing to your suggestion, we added the correct references in the article as following:

Chick, H,; Hume, E.M. The Production in Monkeys of Symptoms closely resembling those of Pellagra, by Prolonged Feeding on a Diet of Low Protein Content. Biochem. J. 1920, 14(2):135-146, doi:10.4088/JCP.v63n0414. (Please see line 395-396 in the revised version)

Comment 6: L98 “[9]” - is correct?

Authors'responses and locations of the revisions: Thanks. Owing to your suggestion, we refered to the peer article (Effects of Low-Protein Diets Supplemented with Branched-Chain Amino Acid on Lipid Metabolism in White Adipose Tissue of Piglets), after heated discussion, we decided to replace reference [9] with (NRC, 2012). (Please see line 101-102 in the revised version)

Comment 7: In table 1, put a space between L-Lysine HCL and (Lys), DL-Methionine and (Met). Threonine a (Thr), L-Tryptopan a (Trp) 

Authors'responses and locations of the revisions: Thanks. Owing to your suggestion, we added spaces between “L-Lysine HCL and (Lys), DL-Methionine and (Met). Threonine a (Thr), L-Tryptopan a (Trp)” in table 1.  (Please see table 1)

Comment 8: L111 without spaces "(250.,"

Authors'responses and locations of the revisions: Owing to your suggestion, we deleted the space before the number "250" in "(250 RCF for 15 min at 4 ℃)".  (Please see line 115 in the revised version)

Comment 9: L131 N (PN) space

Authors'responses and locations of the revisions: Owing to your suggestion, we added a space between N and (PN). (Please see line 137-140 in the revised version)

Comment 10: L132 N (TN) space

Authors'responses and locations of the revisions: Thanks. Owing to your suggestion, we added spaces between N and (TN). (Please see line 137-140 in the revised version)

Comment 11: L133 N (NPN) space 

Authors'responses and locations of the revisions: Thanks. Owing to your suggestion, we added spaces between N and (NPN). (Please see line 137-140 in the revised version)

Comment 12: L146 non-essential (NEAA) and essential (EAA) amino acids flow and apparent total amino acid number (TAA)…. instead of the current form

Authors'responses and locations of the revisions: Thanks. Owing to your suggestion, we changed the original title to "Feces total AA (TAA), essential (EAA) amino acids and non-essential (NEAA) flow amount and apparent total amino acid number (TAA) with or without microbia". (Please see line 152-153 in the revised version)

Comment 13: L151 "digestibility" instead of diestibility

Authors'responses and locations of the revisions: Thanks. Owing to your suggestion, we replaced "Diestibility" with "Digestibility". (Please see line 159 in the revised version)

Comment 14: L156 there are important differences between the M-CP group and the L-CP group, see Table 3 

Authors'responses and locations of the revisions: Thanks. We rewrite this paragraph “......but there were no significant differences in feces TAA, EAA and NEAA flow amount with or without microbia in growing pigs between the 12% CP group and the 15% CP group (P > 0.05).”  (Please see line 157-159 in the revised version)

Comment 15: L161 Total AA (TAA), essentials AA (EAA), nonessential AA flow (NEAA) ... instead of actual form

Authors'responses and locations of the revisions: Thanks. Owing to your suggestion, we changed the original title to " Feces Total AA (TAA), essentials AA (EAA), nonessential AA flow (NEAA) flow (mg/g DM) and apparent digestibility (%) with or without microbial of Total AA (TAA), essentials AA (EAA), nonessential AA flow (NEAA) of growing pigs fed different crude protein levels diets".

Comment 16: L166 for "Ala" H-CP also significantly higher than L-CP and M-CP?

Authors'responses and locations of the revisions: Thanks. Owing to your suggestion, we checked carefully and added "Ala" to the sentence.

The modified sentence as following: Feces Ser, Glu, Ala, Gly, Tyr, Asp, Thr, Cys, Leu and Phe flow amount were different among the 3 experimental groups (P < 0.01) as illustrated in Table 4, i.e. compared with the 15% CP and 12% CP group, higher in the 18% CP group. (Please see line 175-177 in the revised version)

Comment 17: L168 for Ala M-CP and L-CP are significant differences

Authors'responses and locations of the revisions: Thanks. Owing to your suggestion, we carefully checked and restated our results. The modified sentence as following: Feces Ser, Glu, Ala, Gly, Tyr, Asp, Thr, Cys, Leu and Phe flow amount were different among the 3 experimental groups (P < 0.01) as illustrated in Table 4, i.e. compared with the M-CP (15% CP) and L-CP group (12% CP), higher in the H-CP group (18% CP). However, compared with the M-CP group (15% CP), higher in the L-CP group (12% CP) (P < 0.05). (Please see line 174-177 in the revised version)

Comment 18: L170 for "Leu" H-CP is higher than L-CP and M-CP (P < 0.05)

Authors'responses and locations of the revisions: Thanks. Owing to your suggestion, we modified the wrong description in our paper. The modified sentence as following: Feces Leu flow were different among 3 treatments (P < 0.05), i.e. greater in the H-CP group (18% CP) than that in the L-CP (12% CP) and the M-CP group (15% CP) (P < 0.05). (Please see line 182-183 in the revised version)

Comment 19: In tables 4 to 8, add line before dates (the end of the header)

Authors'responses and locations of the revisions: Thanks. Owing to your suggestion, we have perfected the table information of 4-8 and added line before dates. (Please see tables 4 to 8 in the revised version)

Comment 20: L180 M-CP instead of H-CP

Authors'responses and locations of the revisions: Thanks. Owing to your suggestion, we replaced "H-CP" (18% CP) with "M-CP"  (15% CP) . (Please see line 188-189 in the revised version)

Comment 21: L198-202 what is it?

Authors'responses and locations of the revisions: Thanks. Owing to your suggestion, we checked carefully and deleted the error "Fecal Ala and Val flow of growing pigs" left by the last revision. (Please see line 191-192 in the revised version)

Comment 22: L220 "groups H-CP than in groups M-CP and L-CP ...".

Authors'responses and locations of the revisions: Thanks. Owing to your suggestion, we carefully checked and modified the wrong description.

The modified sentence as following: Feces Met digestibility were highly significant different among these 3 experimental groups (P < 0.01), i.e. greatest in 12% CP group, greater in 18% CP group, lowest in the 15% CP group (P < 0.05). (Please see line 206-208 in the revised version)

Comment 23L233 -155 justification, no girdles and widows at the end of the line

Authors'responses and locations of the revisions: Thanks. Owing to your suggestion, we modified the format of the text. (Please see line 152-258 in the revised version)

Comment 24: There is no SEM value for the EAA / NEAA feature in Table 7

Authors'responses and locations of the revisions: Thanks. Owing to your suggestion, we found the SEM of EAA/NEAA in the original data, which is 0.01.  (Please see Table 7)

Comment 25: L266-268 repeating the description from lines 259-260

Authors'responses and locations of the revisions: Thanks. Owing to your suggestion, we carefully checked and deleted the contents of L266-268. (Please see line 259-260 in the revised version)

Comment 26: L271 "Ileu" instead of Leu

Authors'responses and locations of the revisions: Thanks. Owing to your suggestion, we replaced "Leu" with "Ile". (Please see line 262 in the revised version)

Comment 27: L273 the lowest for "Arg" was the L-CP group, not the H-CP group

Answer 27: Thanks. In the original text, we restated the results of Gly and Arg respectively after checking the table information.

The modified sentence as following:     

The microbial Gly and Arg of AA composition in ileum were different among the 3 experimental groups (P < 0.01), i.e. the Gly was the lowest in the H-CP group and the highest in the M-CP group (P < 0.01), the Arg was the lowest in the L-CP group and the highest in the M-CP group (P < 0.01). (Please see line 267 in the revised version)

Comment 28: L280 M-CP not H-CP

Authors'responses and locations of the revisions: Thanks. Owing to your suggestion, we replaced " H-CP(18% CP)" with "M-CP(15% CP)" in the original sentence. (Please see line 276-277 in the revised version)

Comment 29: L281 "It was ....” please delete - repetition

Authors'responses and locations of the revisions: Thanks. Owing to your suggestion, we carefully check and delete "It was...".

The modified sentence as following:

The ileal microbial Cys and Met of AA composition were different among the 3 experimental groups (P < 0.01), i.e. greatest in the 18% CP group, greater in the 12% CP group, lowest in the 15% CP group (P < 0.05). The ileal microbial Leu of AA composition was different among the 3 experimental groups (P < 0.01), i.e. greater in the 15% CP and 18% CP groups than in the 12% CP group (P < 0.05). There were significant differences in feces microbial Leu of AA composition in growing pigs between the 18% CP group and the 15% CP group (P < 0.05). The ileal microbial Phe, EAA and EAA/NEAA of AA composition were different among the 3 experimental groups (P < 0.01), i.e. greatest in the 18% CP group and lowest in the 12% CP group (P < 0.05). The ileal microbial NEAA of AA composition were different among the 3 experimental groups (P < 0.01), i.e. greater in the 12% CP group than that in the 15% CP and 18% CP groups (P < 0.05). There were no significant differences in feces microbial NEAA of AA composition in growing pigs between the 15% CP and 18% CP groups (P > 0.05). There were no significant differences in feces microbial Tyr and Lys of AA composition in growing pigs among 3the 3 experimental groups (P > 0.05). The ileal microbial Arg of AA composition were different among the 3 experimental groups (P < 0.01), i.e. greatest in the 15% CP group greater in the 12% CP group and lowest in the 18% CP group (P < 0.05).

 (Please see line 276-290 in the revised version)

Comment 30: In the references, the short name of the journal, not a printed character, must usually be letters

Authors'responses and locations of the revisions: Thanks. Owing to your suggestion, we have remodified written form of the short name of the journal in the references. (Please see the part of references in the revised version)

Comment 31: References 2-4, 7-9, 14, 16-20, 22, 25-26, 29 include all the authors of the articles, not just the first three.

Authors'responses and locations of the revisions: Thanks. Owing to your suggestion, we checked and revised all references as required. Finally, we listed all the authors of the articles in references 2-4, 7-9, 14, 16-20, 22, 25-26 and 29.  (Please see the part of references in the revised version)

Reviewer 2 Report

The necessary adjustments were made.

line 202: sentence in the air

Tab 7: the SEM of EAA/NEAA is not reported

Author Response

Editors

Animals

Nov 2, 2020

Re: check of “animals-964715”

"Effects of dietary different crude protein levels on fecal crude protein, amino acids flow, amino acid digestibility, fecal and ileal amino acids composition  and microbial ecology in growing pigs ".

Dear Editor:

I would like to thank the editor for reviewing this article in the midst of his busy schedule. Your comprehensive guidance to our work is our way forward. We are very grateful to the external audit experts for their expert opinions, which not only make us clearly aware of our shortcomings, but also is great significant for our future study and work. We have seriously revised the questions raised by the experts and teachers, and if there are any omissions, we will carefully check the amendments again. Finally, I sincerely appreciate all the teachers who have devoted their time to this article. Finally, I offer my sincerest thank you again.

Sincerely yours,

Zhiru Tang Ph.D, Professor,

Zhenguo Yang Ph.D,

Key Laboratory for Bio-feed and Animal Nutrition,

Southwest University, Chongqing 400715, P. R. China

Additional notes 

In order to express the purpose of this experiment clear, We changed the presentation of the 3 experimental groups: replaced " L-CP" with "12% CP", replaced " M-CP" with "15% CP" and replaced "H-CP" with "18% CP".

In addition, we also corrected some errors we found ourselves and highlighted them in red.

Section : "Responses to the Comments by reviewer 2"

Comment 1: line 202: sentence in the air

Authors'responses and locations of the revisions: Thanks. Owing to your suggestion, we carefully examined and rewrite the whole sentence. In addition, we deleted the errors left over from the last modification, such as the short sentences “Fecal Ala and Val flow of growing pigs”.

Comment 2: Tab 7: the SEM of EAA/NEAA is not reported

Answer 2: Thanks. Owing to your suggestion, We missed this data when we revised. Now we have fill 0.01 in Tab 7.